# Causal Algorithm Design: Multi-Resolution KG Extraction and Paper-Code Grounding for ML Algorithms

Eduard-Antonio Zippenfenig[1,2], Andrey Ustyuzhanin[1,2,3,‡]

[1]*Constructor Labs, Campus Ring 1, 28759 Bremen, Germany*

[2]*Constructor University, Campus Ring 1, 28759 Bremen, Germany*

[3]*Institute for Functional Intelligent Materials, National University of Singapore, 4 Science Drive 2, Singapore 117544, Singapore*

## Abstract

We present Causal Algorithm Design, a functional system that uses a single large language model to extract typed causal dependency graphs of machine-learning algorithms from heterogeneous artifacts, research papers (PDF) and reference implementations (Python source), and to align the two extractions against each other. Each algorithm is rendered as a pair of coupled sub-graphs separating training-time from inference-time causality, and at three explicit abstraction levels (micro / meso / macro, 12–25 / 7–10 / 4–7 nodes) produced from the same source under prompt control. Nodes are drawn from a fixed seven-type ontology — input, parameter, mechanism, state, aggregation, decision, output — and emitted as Mermaid, a parseable and renderable surface form that makes downstream automated consumption straightforward. A grounding stage then produces a tabular alignment between the paper-derived and code-derived inference graphs, providing a concrete instrument for inspecting grounding faithfulness of an algorithm description against its executable ground truth. We accompany the system with a small reproducible corpus of 14 paired paper/code artefacts spanning classical ML (e.g., K-means, PCA, Random Forest), transformers, and recent KG-oriented methods, positioning it as a micro-benchmark for cross-modal KG construction and grounding evaluation. A cross-model study (Claude Opus 4.6, Sonnet 4.6, Haiku 4.5) shows that the three abstraction levels form a 84–96% consistent hierarchy across model sizes, with Opus and Sonnet within 1% of each other on Micro→Meso coverage and Haiku trailing by ∼9%.

## Keywords

knowledge graph extraction, causal representation, algorithm grounding, large language models, multi-resolution abstraction

## 1. Introduction

ML algorithms exist in two canonical forms — a paper that communicates intent and high-level mechanics, and code that encodes executable ground truth — and they rarely agree. Papers idealise; code adds numerical safeguards, engineering shortcuts, and API constraints the paper never mentions. The gap is a persistent obstacle to reproduction, audit, and composition.

Knowledge graphs (KGs) could in principle bridge this gap: if both forms are rendered as typed graphs over a common ontology, alignment becomes graph matching rather than open-ended reading. Constructing such graphs has historically required expert annotation [1] or formalism-specific parsers. We use a large language model instead — feeding it the raw PDF or the raw Python file — and ask whether the resulting graphs are consistent, typed, and faithful enough to be useful.

Our system, *Causal Algorithm Design*, makes three commitments: (i) graphs are *causal* (mechanistic dependency, not co-occurrence) [2]; (ii) the same source is rendered at three granularities (micro / meso / macro: 12–25 / 7–10 / 4–7 nodes) by varying only the prompt; (iii) the paper-graph and code-graph are not products in their own right but inputs to a *grounding stage* that aligns them and labels every unmatched code node as a conceptual or implementation gap. We release a 14-algorithm corpus paired with all generated artefacts and verify the multi-resolution hierarchy across three Claude model sizes (Opus 4.6, Sonnet 4.6, Haiku 4.5); generation experiments use Opus 4.6, traced via LangSmith.

*IJCAI 2026: 35th International Joint Conference on Artificial Intelligence, August 16–22, 2026, Bremen, Germany*

✉ andrey.ustyuzhanin@constructor.org (A. Ustyuzhanin)

## 2. Related Work

**KG construction and LLM-driven extraction.**    Automated KG construction from scientific text was historically pipelined IE — NER, relation classification, coreference [3] — later subsumed by joint pretraining objectives [4, 5]. Instruction-tuned LLMs now extract triples zero-shot [6, 7, 8], but this literature targets *relational* graphs in domains such as biomedical text. Our target is a typed *causal* graph over an algorithmic description.

**Algorithm and code understanding; paper–code alignment.**    Program analysis has long produced control- and data-flow graphs from code [9]. Code-aware models such as Code2Vec [10], Codex [11] and StarCoder [12] encode programs as vectors or generate them, but do not extract a structural graph aligned with prose. Reproducibility studies [13] and fact-verification benchmarks [14, 15] document the existence of paper–code drift but provide no structural alignment instrument; PapersWithCode[1] catalogues pairings without a schema. Our grounding stage is a node-level alignment between a paper graph and a code graph over a shared causal ontology.

## 3. System Description

### 3.1. Ingestion

PDFs are sent to the model as base64 documents via the Anthropic document API; Python files are sent as raw text. The same prompt structure handles both. An optional pre-flight YES/NO query (10-token budget) confirms the named algorithm is actually present in the artifact, guarding against multi-method papers.

### 3.2. Diagram Generation

Every diagram is a Mermaid `flowchart TB` block with nodes typed by a fixed seven-element ontology — *input, parameter, mechanism, state, aggregation, decision, output* — enforced through `classDef` colours in the prompt. Each algorithm is decomposed into two coupled sub-graphs: a *training* graph and an *inference* graph, with disjoint node identifiers. Trained parameters re-enter the inference graph only as fixed Parameter nodes, making the phase handoff explicit. Algorithms without a training phase are reinterpreted as *initialisation* and *operational* graphs. Three prompt variants control granularity — micro (12–25 nodes), meso (7–10), macro (4–7) — sharing ontology, colours, and output format. The opening instructions of the micro-level diagram prompt and the grounding prompt are reproduced verbatim in Appendix A; full text for all stages is available in the repository's `prompts/` directory. Figure 1 shows the meso output for Random Forest.

### 3.3. Grounding

The grounding stage aligns the inference sub-graph of the *code-derived* micro diagram (treated as ground truth) against the inference sub-graph of the *paper-derived* micro diagram. The same model is prompted as an SCM expert: every code node is matched to its closest paper node (each paper node consumed at most once), and unmatched code nodes are classified as *conceptual gaps* (omitted causal mechanisms) or *implementation gaps* (engineering details: dtypes, allocations, parallelism). Matched pairs receive Agreement (High/Partial/Low) and Paper-too-abstract (Yes/Partial/No) labels. The output is a 7-column tab-separated table.

---

[1] https://paperswithcode.com

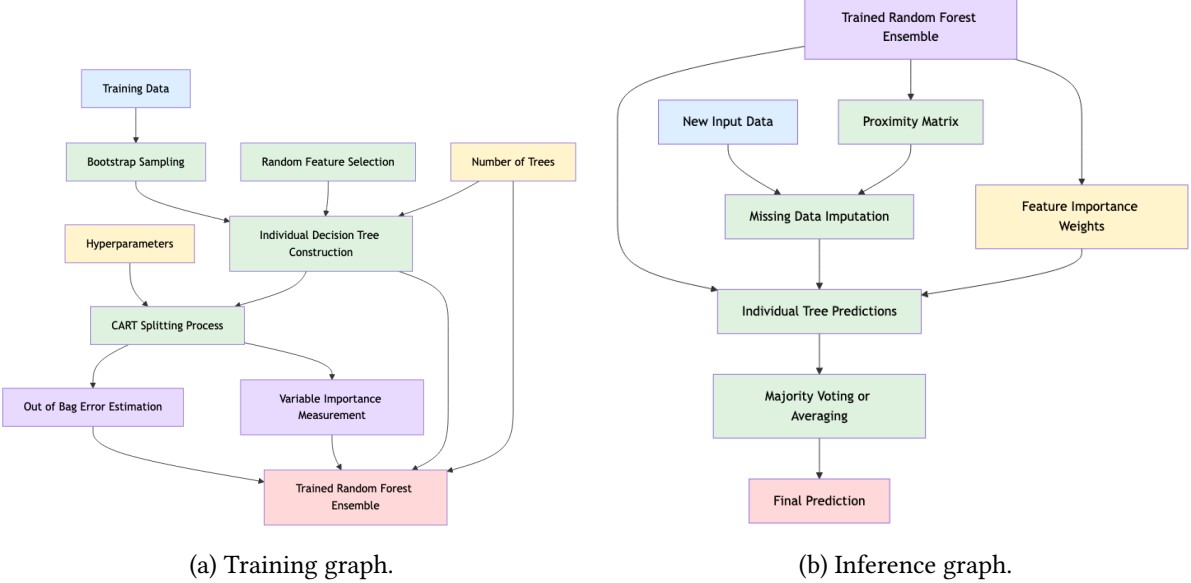

(a) Training graph.            (b) Inference graph.

**Figure 1:** Meso-level causal diagrams for Random Forest, generated from the paper artifact under the same prompt template. The two diagrams share the seven-type node ontology and colour scheme: ▢ Input, ▢ Parameter, ▢ Mechanism, ▢ State, ▢ Aggregation, ▢ Decision, ▢ Output. Trained parameters from a re-enter b only as fixed *Parameter* nodes.

**Table 1**
Algorithm corpus. PDF = research paper provided; PY = Python implementation provided.

| Algorithm | Group | Source |
| --- | --- | --- |
| K-means | Classical ML | PDF + PY |
| PCA | Classical ML | PDF + PY |
| Ridge Regression | Classical ML | PDF + PY |
| Decision Trees | Classical ML | PDF + PY |
| Random Forest | Ensemble | PDF + PY |
| XGBoost | Ensemble | PDF + PY |
| LightGBM | Ensemble | PDF + PY |
| BERT | Transformer | PDF + PY |
| Attention (Transformer) | Transformer | PDF + PY |
| Encoder–Decoder | Transformer | PDF + PY |
| Image Transformer | Transformer | PDF + PY |
| Diffusion Network | Generative | PDF + PY |
| TREK | KG-oriented | PDF + PY |
| Wikontic | KG-oriented | PDF + PY |

## 4. Experimental Corpus

The corpus contains 14 paired (PDF, Python) algorithms (Table 1), stratified into classical ML, ensembles, transformers, generative, and KG-oriented methods. Each algorithm produces three paper diagrams (micro/meso/macro), one code diagram (micro), and one grounding table; all artefacts are version-controlled alongside the source files. The KG-oriented entries (TREK [16], Wikontic) are algorithms *over* knowledge graphs — their causal representations link KG-structured inputs to KG-update outputs — and are the workshop-aligned slice.

## 5. Analysis and Observations

### 5.1. Grounding Gap Distribution

Inspection of the 14 grounding tables reveals a pattern stratified by algorithm family. Classical ML methods exhibit moderate match rates (K-means 6/14, Ridge 7/30, PCA 4/26); their conceptual gaps are rarely about the core procedure but reflect operational branches — for instance the K-means code exposes an *Operation Mode Decision* routing between `predict`, `transform`, `score` that the paper omits. Ensemble methods preserve more of their description (Random Forest 12/30, XGBoost 7/20, LightGBM 12/21 matched), with implementation gaps confined to engineering infrastructure — histogram bin construction, leaf-wise growth, parallel dispatch — that papers mention only obliquely. Transformer methods are the most code-heavy: BERT shows 7 matched out of 37 with 20 implementation gaps for masking, positional encoding, and output projection. Across the corpus, conceptual gaps signal under-described causal mechanisms, while implementation gaps cluster in three repeating buckets (numerical safeguards, parallelism, API plumbing).

### 5.2. Multi-Resolution Consistency

A separate level-comparison stage (prompt: `level_comparison_prompt.txt`) builds Micro→Meso and Meso→Macro mappings for the same algorithm and phase, flags un-mappable nodes as *orphans* and genuinely-multi-parent nodes as *multi-parent* cases, and labels each mapping Full / Partial / Gap. To probe robustness to the judging model, we ran the same stage with three Claude model sizes — Opus 4.6, Sonnet 4.6, Haiku 4.5 — on the same diagrams produced by Opus. Table 2 reports per-algorithm coverage for all three models, summed over training and inference phases.

On Opus, 850/902 (94.2%) micro nodes have a meso parent; 242/252 (96.0%) meso nodes have a macro parent; only $17 + 2$ nodes are multi-parent — the hierarchy is overwhelmingly tree-like, not lattice-like. Orphans cluster on auxiliary mechanisms rather than core flow: Random Forest's proximity-matrix and missing-value-imputation loops, Ridge's Sherman–Morrison residual update, Decision Trees' pruning and impurity variants. The cleanest case is K-means (100%/100%); the worst is Decision Trees (84%/85%), reflecting a paper that already foregrounds engineering details so the meso prompt has nothing left to absorb. Sonnet matches Opus on Micro→Meso (93.6% vs. 94.2%) but trails on Meso→Macro (93.1% vs. 96.0%); Haiku trails both, particularly on Micro→Meso (84.8%), suggesting smaller models do less aggregation work at the finer-grained mapping. Per-algorithm rank ordering is broadly stable (K-means easiest; Decision Trees and Encoder–Decoder hardest), but isolated cells diverge sharply — e.g. Haiku's 58% on Wikontic Micro→Meso. Two model-format hazards surfaced: Sonnet emits "Table N header:" markers that the parser must tolerate, and Haiku occasionally drops header rows entirely (causing the BERT cell to be marked N/A). Robustness to surface-format variation is therefore a real engineering concern. Overall, the qualitative claim that the three resolutions form a hierarchy is quantitatively supported and is robust across model sizes, with a non-negligible auxiliary-edge leak.

### 5.3. Constructive versus Direct Hierarchy Generation

§5.2 measures whether independently-generated micro/meso/macro diagrams (direct generation, also called level comparison) are mutually consistent. A separate question is whether *constructive* generation — building the meso diagram from the micro diagram, and the macro diagram from the meso diagram, via the hierarchy prompts introduced in §3 — produces a more consistent hierarchy in the first place, and at what cost. We compare the two pipelines on the same 14-algorithm corpus using the same Micro→Meso / Meso→Macro coverage metric, plus a structural-outlier count (a node transition between abstraction levels that is inconsistent with the rest of the diagram) and the token cost of each pipeline as measured via LangSmith tracing.

The constructive pipeline wins or ties on every one of the 14 algorithms — 12 outright wins, 2 ties, zero losses (Table 3) — raising mean coverage from 93.7%/93.7% to 99.8%/100.0%. Counting structural

**Table 2**
Multi-resolution coverage across three Claude models (training + inference combined). Columns labelled M→M = Micro→Meso, Me→Ma = Meso→Macro; values are percentages. N/A marks an unparseable model output.

| Algorithm | Opus 4.6 | | Sonnet 4.6 | | Haiku 4.5 | |
|---|---|---|---|---|---|---|
| | M→M | Me→Ma | M→M | Me→Ma | M→M | Me→Ma |
| K-means | 100 | 100 | 90 | 100 | 88 | 100 |
| Random Forest | 95 | 100 | 100 | 89 | 90 | 100 |
| XGBoost | 96 | 100 | 100 | 94 | 71 | 94 |
| LightGBM | 94 | 100 | 91 | 80 | 72 | 94 |
| PCA | 98 | 94 | 99 | 94 | 85 | 94 |
| Ridge Regression | 86 | 93 | 98 | 100 | 93 | 93 |
| Decision Trees | 84 | 85 | 80 | 86 | 84 | 89 |
| Attention | 95 | 90 | 92 | 95 | 95 | 100 |
| BERT | 96 | 100 | 100 | 100 | N/A | N/A |
| Encoder–Decoder | 88 | 89 | 93 | 89 | 94 | 82 |
| Image Transformer | 98 | 100 | 92 | 100 | 94 | 100 |
| Diffusion Network | 97 | 94 | 84 | 100 | 82 | 94 |
| TREK | 94 | 100 | 94 | 95 | 95 | 85 |
| Wikontic | 97 | 100 | 93 | 83 | 58 | 100 |
| **Corpus** | **94.2** | **96.0** | **93.6** | **93.1** | **84.8** | **93.9** |

**Table 3**
Direct versus constructive pipeline coverage on the 14-algorithm corpus. Columns report Micro→Meso (Mi→Me) and Meso→Macro (Me→Ma) coverage as fractions.

| Algorithm | Direct | | Constructive | | Winner |
|---|---|---|---|---|---|
| | Mi→Me | Me→Ma | Mi→Me | Me→Ma | |
| BERT | 1.000 | 1.000 | 1.000 | 1.000 | Tie |
| Wikontic | 0.639 | 0.647 | 1.000 | 1.000 | Constructive |
| Decision Trees | 0.940 | 1.000 | 1.000 | 1.000 | Constructive |
| K-means | 1.000 | 1.000 | 1.000 | 1.000 | Tie |
| LightGBM | 0.950 | 1.000 | 1.000 | 1.000 | Constructive |
| TREK | 0.979 | 0.889 | 1.000 | 1.000 | Constructive |
| PCA | 0.978 | 0.900 | 0.983 | 1.000 | Constructive |
| Ridge Regression | 1.000 | 0.944 | 1.000 | 1.000 | Constructive |
| XGBoost | 0.932 | 0.895 | 1.000 | 1.000 | Constructive |
| Attention | 0.913 | 1.000 | 1.000 | 1.000 | Constructive |
| Diffusion Network | 0.969 | 0.944 | 1.000 | 1.000 | Constructive |
| Encoder–Decoder | 0.885 | 0.895 | 0.982 | 1.000 | Constructive |
| Image Transformer | 0.984 | 1.000 | 1.000 | 1.000 | Constructive |
| Random Forest | 0.950 | 1.000 | 1.000 | 1.000 | Constructive |
| **Corpus** | **0.937** | **0.937** | **0.998** | **1.000** | |

outliers across the whole corpus rather than per-algorithm rates makes the gap starker: 70 outliers under direct generation versus 2 under constructive generation, a $35\times$ reduction. The constructive pipeline is also *cheaper* than generating each level independently, since the meso and macro prompts consume the already-extracted micro diagram instead of re-reading the source artifact at every level. Hierarchical derivation is therefore not only a cleaner causal story — a meso node is by construction an aggregate of specific micro nodes, rather than an independently-judged abstraction that happens to resemble one — but is simultaneously more consistent and less expensive than the direct alternative used in §5.2.

# 6. Limitations

**No quantitative grounding evaluation.** Precision/recall against a human-annotated gold standard remains future work. **LLM-as-judge bias.** The same model builds graphs and aligns them; an external judge is required for independent faithfulness verification. **Mermaid is not a formal KG.** No URIs, RDF, or SPARQL; the ontology is implicit. Lifting to OWL/RDF-star is a downstream concern. **Single-model dependence.** All experiments use Claude Opus 4.6; no cross-family ablation. **Small, ML-skewed corpus.** 14 algorithms; coverage of systems, optimisation, and domain methods absent. **Paper–code drift conflated with hallucination [13].** The grounding table cannot currently distinguish the two failure modes. **Phase template is brittle.** Online learning, unsupervised pipelines, and agentic loops are forced into the train/infer split; the init/operational fallback is heuristic.

# 7. Conclusion

*Causal Algorithm Design* renders ML algorithms as paired typed causal graphs from both papers and code, at three abstraction levels, and aligns the two via a structured grounding stage. The system makes four contributions to the workshop's themes:

**C1 Multimodal causal KG extraction.** A single prompt structure produces typed causal graphs from both a PDF and its reference implementation, populating a seven-type ontology without modality-specific parsers.

**C2 Cross-modal grounding** as paper–code alignment, classifying every code node as matched, conceptual gap, or implementation gap — instantiating the workshop's *fact-verification* and *explainability with provenance* topics.

**C3 Multi-resolution hierarchy robust across model sizes.** Three Claude models (Opus 4.6, Sonnet 4.6, Haiku 4.5) independently judge the same 14-algorithm benchmark and produce broadly consistent corpus coverage — $94.2\%/96.0\%, 93.6\%/93.1\%, 84.8\%/93.9\%$ Micro→Meso/Meso→Macro respectively (§5.2) — with stable per-algorithm rank ordering and smaller models lagging only on the finer-grained mapping. The hierarchy is therefore not an artefact of any single model's biases or an accidental consequence of independent generations.

**C4 Mermaid as a constrained-generation target**: output is parseable, renderable, and diffable, with the ontology and node-count limits acting as structural guardrails on the LLM.

The corpus of 14 paired paper/code artefacts together with the pipeline scripts constitutes a reproducible micro-benchmark for cross-modal KG construction and grounding evaluation, supporting future quantitative faithfulness studies.

## Declaration on Generative AI

This paper describes a system whose primary outputs are produced by generative AI. Claude Opus 4.6 (`claude-opus-4-6`) is invoked by the released pipeline scripts to (i) generate every causal Mermaid diagram (training and inference, at all three resolutions) for the 14 algorithms in the corpus, (ii) produce each paper–code grounding table, and (iii) gate the pre-flight algorithm-presence check. Claude Sonnet 4.6 and Claude Haiku 4.5 were additionally used as judges in the multi-model robustness study reported in §5.2. The diagrams in Figure 1, the gap counts in §5.2, and the coverage figures in Table 2 are direct, unedited outputs of the named models from the published prompts.

During manuscript preparation, the authors additionally used Claude (Opus 4.7, via Claude Code) as a writing and engineering assistant to draft prose, restructure sections under the page-budget constraint, populate the BibTeX file, and apply LaTeX edits. Every numerical claim was verified against the underlying `xlsx` outputs and Python scripts before inclusion; bibliographic entries were independently checked against the ACL Anthology, arXiv, IEEE Xplore, and publisher pages. The authors reviewed all generated content and take full responsibility for the publication.

## Acknowledgement

This project was supported by the Ministry of Education, Singapore, under its Research Centre of Excellence award to the Institute for Functional Intelligence Materials (project No. EDUNC-33-18-279-V12) and by the National Research Foundation, Singapore under its AI Singapore Programme (AISG Award No: AISG3-RP-2022-028).

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

# A. Prompt Excerpts

The following reproduces the opening instructions of the two core prompts verbatim. Full prompts for all stages (micro, meso, macro, hierarchy variants, level comparison, method comparison) are in the `prompts/` directory of the accompanying repository.

## Micro-level diagram generation (`micro.txt`, excerpt)

```
You are a research-grade analytical assistant specialized in algorithm
interpretation, computational causality, mechanistic analysis, and formal
diagram design.

Your task is to read an attached document that describes an algorithm,
computational method, workflow, or system process, and convert it into two
separate high-detail causal influence diagrams using Mermaid.

The attached document is the only source of truth. Do not invent mechanisms
or variables that are not supported by the document.

GOAL
Produce mechanistic causal abstractions that reveal how inputs, parameters,
internal states, and transformations jointly produce outputs.

CRITICAL REQUIREMENT: TWO SEPARATE CAUSAL DIAGRAMS
You must produce two completely independent causal diagrams representing two
different causal regimes of the algorithm. They must be implemented as two
separate Mermaid flowcharts. Do NOT place both diagrams inside the same
flowchart. The diagrams must not share nodes, identifiers, or structures.

DIAGRAM 1 - TRAINING / LEARNING CAUSAL MODEL
This diagram represents the causal process through which parameters, models,
or internal structures are learned, estimated, or configured.
The output of this diagram must be the trained model, parameter set, or
configured system.

DIAGRAM 2 - INFERENCE / EXECUTION CAUSAL MODEL
This diagram represents the causal process through which the finalized
algorithm produces outputs from inputs. The trained parameters produced in
the training diagram must appear here only as fixed inputs.

IF THE ALGORITHM DOES NOT INVOLVE LEARNING
Interpret the diagrams as:
  Diagram 1: Initialization, configuration, or preprocessing pipeline.
  Diagram 2: Operational execution pipeline.
```

## Grounding prompt (`grounding_prompt.txt`, excerpt)

```
You are an expert in causal inference and causal representation learning,
familiar with the framework of Judea Pearl and structural causal models (SCMs).

You are given two causal representations of the same algorithm:
- REPRESENTATION A: Code-based causal diagram - derived directly from
  implementation. This is the GROUND TRUTH. Inference phase flowchart only.
- REPRESENTATION B: Paper-based causal diagram - derived from the research
  paper describing the algorithm. Inference phase flowchart only.

TASK
Step 1 - Build a one-to-one mapping between code nodes and paper nodes.
Go through every node in REPRESENTATION A (code). For each code node, find
the single best matching node in REPRESENTATION B (paper). Each paper node
may only be matched to ONE code node. A match is valid if the two nodes
```

represent the same causal concept, even if described differently. If no paper
node is available, write "Not present" in the Paper equivalent column.

Step 2 - Classify each code node.
If matched to a paper node -> Gap Type = No gap
If NOT matched (Paper equivalent = Not present) -> classify as either:
  Conceptual gap: the node reflects a core causal mechanism of the algorithm
  Implementation gap: the node is a code-level detail not expected in a paper
  description (e.g. dtype handling, API calls, memory allocation, numerical
  stability tricks, parallelism settings)

OUTPUT FORMAT
Return ONLY a tab-separated table with no additional text before or after it.
Header: Code Node  Node Type  Diagram  Paper equivalent  Agreement  Gap Type