# OpenReview forum: "Causal Algorithm Design: Multi-Resolution KG Extraction and Paper-Code Grounding for ML Algorithms"
_ijcai.org/IJCAI-ECAI/2026/Workshop/GENAIK-NORA — IJCAI-ECAI 2026 Joint Workshop on GENAIK and NORA_

### Official Review · Reviewer_hbfu · 2026-06-02
**a clear approach for algorithms from paper and code, but where can it be used to?**

**Rating:** 4
**Confidence:** 3

**Review:**

This paper suggest causal algorithm design that uses a single llm to extract typed causal dependency graphs of machine-learning algorithms from heteroneous artifacts, research papers and reference implementations. This study assumes paper and code are two canonical forms of how ML algorithms exist and claim that KGs are bridge for their gap.

Authors thereby first send PDFs to document API and get base64 data, generate diagram with Mermaid flowchart typed by seven-element ontology, and groundthe inference subgraph of the code-derived micro diagram against the inference subgraph of the paper-derived micro diagram. Tested with 14 algorithms including K-means, XGBoost, BERT etc. authors find out that the trend of grounding differs by group and multi-resolution coverage differs across the LLM models.

I read the paper interesting and I like the idea that algorithms themselves can be taken into account as a target of analysis with KG, though authors admit that mermaid is not necessarily a formal KG. I think it is not necessarily the weakness of the study.

However, I wonder what the impact and application of this research output would be. Authors state in the conclusion that the proposed CAD renders algorithms as paired typed graphs from both papers and code -- in what aspect of science would the proposed scheme contribute to? I hope there is any specific use cases (at least visualization of KGs for an algorithm) and how such abstractization helps 1) quantification of information, 2) setting further research direction, 3) efficiently formalize and organize the information of algorithm-related data, etc.

I think the paper fits with the workshop, but further development may have to take place for its extension.

---

### Official Review · Reviewer_bUs6 · 2026-06-06
**Causal Algorithm Design: Multi-Resolution KG Extraction and Paper-Code Grounding for ML Algorithms**

**Rating:** 7
**Confidence:** 3

**Review:**

The paper presents a functional pipeline that prompts a single LLM (Claude) to extract typed dependency graphs from two artifacts of the same ML algorithm. The two sources are its paper (PDF) and a reference Python implementation. Each algorithm is rendered as a pair of coupled sub-graphs (training-time / inference-time), at three node-count-controlled resolutions (micro 12–25 / meso 7–10 / macro 4–7 nodes), over a fixed seven-type ontology (input, parameter, mechanism, state, aggregation, decision, output), emitted as Mermaid. A grounding stage matches each code-graph node to a paper-graph node and labels unmatched code nodes as conceptual or implementation gaps. Deliverables: a 14-algorithm corpus of paired artifacts and a cross-model consistency study (Opus 4.6, Sonnet 4.6, Haiku 4.5) reporting 84–96% Micro→Meso / Meso→Macro coverage. Clear, well-structured writing with an accurate abstract and a clean conceptual model (seven-type ontology, train/infer coupling, three resolutions). Table 2 compares only three Claude models to each other; there is no external gold standard or non-LLM comparison. A baseline would clarify not just correctness but the LLM's value over a parser or manual annotation especially its cost and scalability

---

### Official Review · Reviewer_U9gP · 2026-06-08
**A tool for KG extraction**

**Rating:** 6
**Confidence:** 2

**Review:**

The paper presents a system using the Anthropic document API.  It derives causal dependency graphs from algorithms and then aligns them.   It actually produces two graphs and aligns them; in a sense this is the interesting contribution.   It comes with a very small corpus.   I didn't come away with a solid understanding of what the output causal graphs would be used for, or how mechanistic interpretability is used.  The prompt wasn't specified, and I have to wonder whether there was a theoretical analysis that the authors constructed while working on this that is somehow suppressed in the paper.

Overall, it looks possibly promising for others in the KG field.  But I can't say it's a 'must read' for the area.

The fact that it's a student paper at a workshop also suggests that it would be good to have it.

---

### Official Review · Reviewer_LGg4 · 2026-06-09

**Rating:** 5
**Confidence:** 4

**Review:**

# Strengths
1. The paper addresses a real and important issue: research papers and reference implementations often diverge in ways that matter for reproducibility, auditability, and scientific understanding. Framing paper–code drift as a graph alignment problem is a useful and potentially impactful direction for the NLP, scientific-document understanding, and knowledge-graph communities.
2. The use of typed nodes, separate training and inference graphs, and explicit abstraction levels makes the generated artifacts easier to inspect than unstructured LLM summaries. The fixed seven-type ontology provides a simple common schema across papers and code, and Mermaid offers a practical parseable/renderable surface form.
3. Generating micro, meso, and macro graphs from the same source is an interesting way to study abstraction in algorithm descriptions. The paper’s attempt to check whether lower-level nodes map cleanly into higher-level nodes is a useful first step toward validating hierarchical graph representations.

# Weaknesses
1. The paper’s most important claim is that the extracted graphs are useful for grounding paper descriptions against code, but there is no human-annotated gold standard for either the graphs or the paper–code alignments. As a result, the reported match rates and gap categories cannot be interpreted as precision, recall, or accuracy. The paper currently shows that the pipeline produces structured artifacts, but not that those artifacts are correct.
2. The paper relies heavily on Claude models: Opus generates the diagrams and grounding tables, while Opus/Sonnet/Haiku are used for the multi-resolution judging analysis. This leaves substantial room for self-consistency bias. A same-family judge may reward representations that match the prompting style rather than representations that are semantically faithful to the paper or code.
3. The paper treats the code-derived micro inference graph as executable ground truth, but that graph is itself an LLM extraction. It may omit important control flow, hallucinate causal mechanisms, or misinterpret implementation details. Without comparison to static analysis, dynamic traces, human code review, or manually constructed program graphs, it is unclear whether gaps between paper and code reflect real paper–code drift or errors in the LLM’s code understanding.
4. Fourteen algorithms are useful for a pilot benchmark, but they are not enough to establish generality. Many are standard algorithms that LLMs may already know well, which raises concerns about memorization or reliance on prior knowledge rather than artifact-grounded extraction. The two KG-oriented examples are especially few if the intended contribution is positioned around KG-oriented methods.

---

### Decision · Program_Chairs · 2026-06-10

Accept